# Multi-stakeholder perspectives on gender-based violence in marital relationships: A qualitative study in rural India

**Soumya Singh[1], Aneel Brar[1,2,3,4], Urvita Bhatia[1,5], Devika Gupta[1,5], Abhijit Nadkarni[1,5*]**

1 Addictions and Related Research Group, Sangath, Porvorim, Goa, India, 2 Mata Jai Kaur Maternal and Child Health Centre, Sri Ganganagar, Rajasthan, India, 3 School of Anthropology and Museum Ethnography, University of Oxford, Oxford, United Kingdom, 4 Brain and Mind Institute, Aga Khan University, Nairobi, Kenya, 5 Centre for Global Mental Health (CGMH), Department of Population Health, London School of Hygiene & Tropical Medicine, London, United Kingdom

* abhijit.nadkarni@lshtm.ac.uk

## Abstract

In India, an estimated 41% of women report experiencing domestic violence in their lifetime and up to 28% report violence during pregnancy. Our study aims to understand local perceptions/attitudes towards GBV and lived experiences of GBVvictimisation and perpetration in marital relationships in rural Rajasthan. We conducted a focus group discussion with community health workers (n = 7) and interviews with young married men (n = 17) and women (n = 21), and local government officials (n = 2). The data was analysed using thematic analysis with triangulation conducted across methods and researchers. Our findings reveal that women experience multiform violence, including physical, sexual, and economic violence. In the continuum of GBV, violence stems from everyday conditions of disempowerment produced through early marriage and reproduction, norms of hegemonic masculinity, and community norms that normalise violence against women. These conditions of disempowerment lead to women's isolation through control of mobility and surveillance through technology, setting the stage for further violence to occur. Violence prevention efforts in the region must follow a multi-level approach to reduce child marriage through community education and improved enforcement of legislation, enhance reporting and identification alongside help-seeking behaviours through formal or informal channels, transform rigid gender norms, and leverage key actors in the family system such as the in-laws or matchmakers as potential agents of change.

## Introduction

India has one of the highest rates of marriage in the world. By age 45–49 years, ninety-nine percent of women in India are "ever married" [1]. Further, despite laws prohibiting child marriage, India has a high proportion of child marriage [2,3]. According to

**Data availability statement:** The data reported in this paper contain sensitive information from human participants and cannot be publicly shared due to ethical and confidentiality restrictions. De-identified data may be made available to qualified researchers upon reasonable request. Requests should be directed to irb@sangath.in.

**Funding:** This study was funded through a grant (MAV-027562-68049 to AB) from Harvard Medical School Center for Global Health Delivery. The funders had no role in study design, data collection and analysis, decision to publish, or preparation of the manuscript.

**Competing interests:** The authors have declared that no competing interests exist.

the National Family Health Survey- 5 (2019–2021), 38% of women between the ages of 20–49 years report getting married before they turn 18 [1]. Marriage practices vary widely among ethnic and religious groups, however a common feature in many parts of India is that a married woman is considered to belong to the married household and is no longer a member of the birth family unit [4,5]. Early marriage along with the uncertainty and upheaval associated with marriage render young women powerless and vulnerable [6,7]. Early marriage is linked with early sexual activity and childrearing and psychological and social isolation that puts young women at high risk of psychological, sexual, and emotional violence [8]. Marriage in many parts of India is not only a social institution but a key site through which women's personhood, care, and belonging are understood and organized [9,10]. These same kinship structures, while providing social support, can also produce conditions of vulnerability, particularly for young married women navigating new and often hierarchical relational environments—in other words, marriage can become a platform for violence [11].

In India, an estimated 29% of women aged 18–49 have experienced physical violence since the age of 15 and 32% of ever married women aged 18–49 have experienced spousal violence [1]. In 2023, 38 cases of honour killings were reported, though sources indicate that honour-based killings are largely under-reported [12,13]. Dowry-related violence is also commonly implicated. According to the National Crime Records Bureau (NCRB), a staggering 6156 dowry deaths cases were reported to the police in 2023. The majority of dowry-related deaths are due to 'bride burning' where a woman is doused with flammable liquid and set on fire [14]. The reported rates of cruelty by a husband or his relatives rose by 53%, from 18.5% in 2001 to 28.3% in 2018 per 100,000 women between the ages 15–49 years [15,16]. Further, in 2023, a total of 29,670 rape cases were reported, with 852 rape victims reported under the age of 18 [16]. Despite the existence of several legislative acts such as The Protection of Women from Domestic Violence Act, 2005 [17], Dowry Prohibition Act, 1961 [18]; and The Protection of Children from Sexual Offences (POCSO) Act, 2012 [3], gender-based violence remains pervasive. National statistics are likely to be underestimates due to underreporting caused by stigmatisation of help-seeking for domestic violence, lack of standardised measurements, and inadequate service provider responses [7]. While these figures underscore the scale of gender-based violence, such violence is not only an individual or behavioural phenomenon but is embedded within broader patriarchal kinship and social structures that shape gendered expectations, authority, and control within marriage.

As a result of gender-based violence (GBV), women experience a range of deleterious short- and long-term physical and mental health consequences including physical injury, adverse sexual and reproductive outcomes such as unintended pregnancies and sexually-transmitted infections, depression, anxiety, suicidal behaviour, self-harm, and feelings of guilt, shame, and poor self-esteem [6,19,20]. Furthermore, in rural India, violence is not considered by many to be a problem needing resolution [7]. Women often fear being placed at risk for further violence or abandonment by their husbands if they disclose GBV experiences, creating an additional barrier to help-seeking [21]. These dynamics are not uniform, but are shaped by intersecting

factors such as caste, class, age at marriage, and rural context, which together influence both the experience of violence and the possibilities for seeking support [22].

Understanding these socially embedded dynamics of violence is critical for designing interventions that engage with, rather than operate outside of, the relational and normative contexts in which violence occurs. The engagement of men is being increasingly identified as a key component of interventions that aim to change harmful norms of masculinity, including through the use of positive role modelling from older men in positions of authority [23,24]. Though potentially promising interventions such as *Parivartan* [23], which leverages older men as agents of change, have been implemented in India, there is very limited understanding of community buy-in for such male-led programmes.

Therefore, the objectives of our study were to (1) examine the perceptions and experiences of GBV from a socio-cultural lens, within the context of marriage, among survivors, perpetrators, and community members in rural India; and (2) use the findings to inform the development of a male-led GBV-prevention program. This paper describes our findings from the formative research (objective 1) used to inform the prevention program (objective 2). Our study uses a qualitative phenomenological approach to understand lived experiences of GBV and help-seeking, community perceptions towards GBV prevention, and possible normative explanations for the perpetration of violence in marital relationships. We defined GBV as "any act of gender-based violence that results in, or is likely to result in, physical, sexual, or mental harm or suffering to women, including threats of such acts, coercion or arbitrary deprivation of liberty, whether occurring in public or in private life" [25]. Our definition, thereby, captured all forms of violence including physical, sexual, economic, and emotional violence perpetrated by married partners.

## Methods

### Ethics statement

The study received ethics approval from the Institutional Review Boards (IRB) of the Harvard Faculty of Medicine (IRB19-1326) and Sangath (AN_2019_54).

### Study design

Our qualitative study was nested in a wider formative research project aimed at informing the development of an intervention for GBV. We conducted semi-structured interviews (SSIs) and focus group discussions (FGDs) with key community stakeholders.

### Setting

The study was conducted in rural Sri Ganganagar, a district in Northern Rajasthan, India. Rajasthan is an economically disadvantaged state wherein 28.3% of rural women between the ages 20–24 are married by the age of 18 years, and 4.2% rural girls between the ages of 15–19 are mothers or pregnant [1]. Owing to its agricultural output, Sri Ganganagar is economically relatively better off compared to the rest of Rajasthan; however, it experiences higher levels of economic inequality along caste lines [26,27], and as a result of unequal farm land and irrigated land distribution. Households in the upper 20 percent have an average annual income of Rs. 827,477, while those in the lower 20 percent have an average annual income of Rs. 92,427 [28]. In Rajasthan, 24% of ever-married women aged 18–49 have experienced physical violence and/or sexual violence and violence is most commonly perpetrated by husbands [1].

### Sample

We used purposive sampling to recruit participants from the following stakeholder populations of interest:

(a) young (18–30 years) married women (n = 21) and young married men (n = 17) to understand the lived experiences of and/or explanatory models of GBV perpetration and victimisation. We did not exclusively include survivors and

perpetrators only. Our underlying assumption, based on our experience delivering a community-based perinatal mental health intervention, was that GBV is relatively common within the context, and hence, the inclusion criteria was open to including any married man or woman. We purposely sampled for men and women from both low-income and high-income families to ensure that we capture diversity in attitudes towards and experiences of violence;

(b) Government officials (n = 2) working at the local health centre and village governing body (*panchayat*) to understand the experiences of victimisation and help-seeking at the community level; and

(c) Community health workers (CHWs) such as ASHA workers, *Anganwadis*, and *Sathins* who are at the frontline of responding to GBV (n = 7) to understand their perceptions of violence perpetration and victimisation, as well as identifying possible avenues for prevention within the health system. Accredited Social Health Activists (ASHAs) are community health workers employed under India's National Rural Health Mission. Anganwadi workers are workers in the rural childcare centre set up as part of the Integrated Child Development Services program. *Sathins* are community volunteer workers initially intended to engage with survivors of domestic violence. *Sathins* are trained on details of women-centric schemes such as Mahila Shakti Kendra; community outreach and leadership; legal protections and rights and opportunities for women; and improving dialogue and collaboration between women groups.

Young married women and men were recruited using posters and word-of-mouth advertising through existing community networks. CHWs and government officials were identified through their association as stakeholders in existing programs or through community networks.

## Data collection

We conduced SSIs with the young married women and men, and government officials; and one FGD with the CHWs. The duration of recruitment was 25 November 2019 (Start) to 17 February 2020 (End). After obtaining signed, informed consent, interviews were conducted in the participant's home or workspace if there was adequate space and privacy. Otherwise, interviews were conducted in private spaces outside the home, but within the community, including local temples, schools, or clinics. The FGD was conducted in a private room at a local not-for-profit organisation. Data were collected by trained researchers who were supervised for quality by the co-investigator (AB) using audio recordings of the SSIs and FGD. Socio-demographic information was collected including age, village of residence, caste, religion, level of education, income, and age of marriage. The SSIs and FGDs were conducted in the local language (Hindi, Punjabi, or Rajasthani/Bagri) in which the participants were most comfortable. We conducted informal community meetings to field test vernacular words and phrases used in the interview guides. These meetings helped assess whether the words and phrases were stigmatizing, conceptually similar to the English usage or elicited any bias or cultural resistance. The interview guides covered the following domains:

(a) Gender norms: Understanding normative expectations from married men and women, and their roles and responsibilities.

(b) Community perceptions of GBV: Understanding perceived reasons for GBV perpetration and victimisation as well as the perceived nature and severity of GBV in their own community.

(c) Subjective experiences of GBV perpetration and victimisation: Understanding the experiences of sexual, financial, physical, emotional violence victimisation or perpetration.

Audio recordings of SSIs and FGDs were transcribed verbatim, translated to English, and de-identified. Translation and transcription were conducted in a single step by an external translator. The accuracy of translations was ensured by the authors fluent in the local languages and English.

## Data analysis

We used a thematic analysis approach [29] to inductively derive codes, themes and categories, from the data through the following five stages. In the first stage, all coders read and reread the transcripts 1–2 times to become familiarised with the data. Following this, two transcripts were open-coded by four researchers (SS, PP, AB, AD), resulting in four separate lists of codes and accompanying descriptions. The coders then met to reach a consensus on the codes, which led to the development of a codebook. The remaining transcripts were coded by the same four researchers. The codebook was continuously updated to incorporate new codes or revise existing codes. The themes were discussed in parallel with codebook iteration. A final meeting was held to discuss emergent themes with all team members. Coding was done using NVivo version 12 [30].

## Ethical considerations

We created a directory of the local government, non-governmental, and private resources, and support systems available for those experiencing GBV. Information pamphlets with these support systems, along with more generic health services, were created and translated into Hindi for distribution to all participants and any household members or community members. Providing information on additional services helped disguise violence-specific services, reducing the potential for research engagement to cause discomfort or induce violence from the participant's household or family members. We developed a number of tools and procedures, informed by previous practice, to safeguard participants from potential risks if household or community members were to find that the study centrally focuses on gender-based violence, which could lead to conflict or violence. These included developing a backstory for researcher and creating a dummy questionnaire with questions unrelated to GBV and DV, in the case that a spouse or a family member who is a perpetrator of violence, walks into the interview; and identifying private spaces outside the household to conduct SSIs and the FGD. The dummy questionnaire was not used as a data collection tool – it served as a risk mitigation strategy [31]. Finally, given the sensitive nature of the study and potential for harm if participants were to speak to each other or community members about their experiences of GBV victimisation and perpetration, we only interviewed one member of a household even if there were two or more eligible individuals, and we only interviewed either young married men or young married women in a given village. Debriefing and referrals were provided to participating individuals in any event of distress. To prevent vicarious trauma among research team members, daily debriefs were conducted to focus on reflections and share resources for self-care.

## Results

Socio-demographic details of the young married women and men, and CHWs are given in Table 1.

We organised emerging themes into three inter-related categories. These categories are placed on a continuum ranging from shared experiences of personal, social, and structural conditions of disempowerment that lead to control of and violence towards women. The themes are indicated in Table 2.

### Theme 1. Conditions of disempowerment

Three themes emerged around structures, norms and practices that subordinate or disempower women, therefore setting the stage for isolation, control, and GBV. These include a) norms around marriage, sexuality and reproduction, b) norms of masculinity, and c) normalisation of violence.

**Norms around marriage, sexuality, and reproduction.** Certain norms around marriage and sexuality disempowered women and created conditions that permit GBV. These included expectations and responsibilities placed on a wife, the fulfilment of which can deem her to be a "good woman" who contributes to creating a "good family". Most men respondents stated that women are expected to be solely responsible for household work and reproduction. Similarly,

**Table 1. Sociodemographic characteristics of participants.**

| Variable | Young Women | Young Men | Community Health Workers | Local Government Officials |
|---|---|---|---|---|
| | N = 21 (45%) | N = 17 (36%) | N = 7 (15%) | N = 2 (4%) |
| | N (%) | N (%) | N (%) | N (%) |
| **Mean age in years** (SD) | 25 (2.48) | 27 (4.54) | 47 (11.23) | 55 (9.89) |
| **Mean age of spouse in years** (SD) | 28 (2.61) | 24 (3.93) | NA | NA |
| **Religion** | | | | |
| Hindu | 11 (52) | 5 (29) | 3 (43) | 1 (50) |
| Muslim | 1 (5) | 0 (0) | 0 (0) | 0 (0) |
| Sikh | 9 (43) | 12 (71) | 4 (57) | 1 (50) |
| **Caste** | | | NA | NA |
| Scheduled Caste (SC) | 5 (24) | 6 (35) | | |
| Scheduled Tribe (ST) | 6 (28) | 0 (0) | | |
| Other Backward Castes (OBC) | 9 (43) | 7 (41) | | |
| General | 1 (5) | 4 (24) | | |
| **Level of Education** | | | | |
| No education | 1 (5) | 0 (0) | 0 (0) | 0 (0) |
| Primary | 3 (14) | 0 (0) | 0 (0) | 0 (0) |
| Secondary | 7 (33) | 8 (47) | 4 (57) | 0 (0) |
| Senior Secondary | 5 (24) | 1 (6) | 1 (14) | 0 (0) |
| Undergraduate | 3 (14) | 7 (41) | 2 (29) | 0 (0) |
| Postgraduate | 2 (10) | 1 (6) | 0 (0) | 2 (100) |
| **Occupation** | | | NA | NA |
| Not working outside of home | 17 (80) | 3 (18) | | |
| Daily wage labour | 0 (0) | 3 (18) | | |
| Farming | 2 (10) | 5 (29) | | |
| Other (government jobs, private sector jobs and/or self-owned business) | 2 (10) | 6 (35) | | |
| **Average Monthly Family Income in Indian Rupees** | 17,262 | 32,550 | NA | NA |
| **Type of Marriage** | | | NA | NA |
| Arranged Marriage | 19 (90) | 15 (88) | | |
| Love Marriage | 1 (5) | 1 (6) | | |
| Semi-arranged Marriage | 1 (5) | 1 (6) | | |
| **Average Duration of Marriage in years** (SD) | 6 (5) | 4 (2.25) | NA | NA |
| **Type of House** | | | NA | NA |
| Kuccha | 4 (19) | 1 (7) | | |
| Pucca | 12 (57) | 10 (60) | | |
| Semi-Pucca | 5 (24) | 6 (33) | | |

*NA = Not Applicable. Scheduled Castes (SC) and Scheduled Tribes (ST) are groups of people recognized by the Indian constitution as among the most marginalized and disadvantaged in India. Other Backward Castes (OBCs) include groups of people recognized as "other socially and educationally backward" classes. Love marriage is defined as one where the partners select each other, with or without the consent of families. Semi-arranged marriage is defined as one where partners select each other, with the consent of families. Arranged marriage is one that is solely driven by families. Kuccha house is defined as a house that does not have a permanent structure. The walls are often made of unburnt bricks, stone, or mud. Pucca house is defined as a house that is designed to be solid and permanent. The walls are made of durable materials such as burnt brick and cement. Semi-pucca house are defined as houses that are a mix of Kuccha (commonly a temporary roof) and Pucca elements (commonly stable walls).*

**Table 2. Thematic representation of GBV in rural Rajasthan.**

| Category 1: Conditions of disempowerment | |
|---|---|
| **Themes** | **Sub-themes** |
| Norms around marriage, sexuality and reproduction | Son preference |
| | Subservient and modest |
| | Cohesive without complaint |
| | Marriage into the unknown |
| | Sexual entitlement or obligation |
| Normalisation of violence | Good and bad victims |
| | Violence as inevitable |
| Norms of masculinity | Men as caregivers |
| | Alcohol use |
| | Dominance |
| | Sexual insecurity |
| **Category 2: Control** | |
| **Themes** | **Sub-themes** |
| Isolation & control | Monitoring modesty |
| | Monitoring social interactions |
| | Technological control |
| | Monitoring movement |
| **Category 3: Experience of violence** | |
| **Themes** | **Sub-themes** |
| Experience of GBV | Sexual |
| | Physical |
| | Emotional |
| | Financial |
| Help seeking | Duty towards family |
| | Negative experience of disclosure |
| | High threshold for violence within marriage |
| Coping | De-escalating & preserving relationships |
| | Agency and support |

many women expressed that fulfilling these normative expectations was their responsibility. At the same time, women often revealed dissatisfaction with their life after marriage and the burden of chores being a source of family conflict. One woman stated, *"Earlier, there were not many responsibilities. There is no issue if we do or don't work at parent's home, but here, we have to do it. There is also the responsibility of the children now. Many changes occur after marriage. Life before marriage with parents is very good while it is not that good after marriage."* Many female participants reported having to learn marital roles and responsibilities at a very young age (as young as 15), including feeling obligated to have sex with their husbands. Fewer men described feeling unhappy with an early marriage. One man stated, *"I wanted to have a job first. After marriage, it becomes more restricted; you can't go anywhere, (you) can't do what you want to do."* The key sub-themes within this theme include a) marriage into the unknown b) subservient and modest: what a "good" woman is, c) cohesive without complaint: what a "good" woman should do, d) sexual entitlement or obligation, and e) son preference.

In the study setting, most marriages are fixed by families and mediated by matchmakers known as *bicholas* and most women do not know or meet their partners before marriage. Marriage for most women was an entry into the unknown, arranged by families and mediated by matchmakers (bicholas). Crucial information about the husband — such as alcohol use, health problems, or existing romantic relationships — was often withheld until after the marriage.

*"When my husband drinks alcohol at night, he argues with me after that, otherwise, he does not argue normally. He says, 'I did not start drinking just now (during the marriage), I have been drinking from the beginning (since before marriage)'. But nobody had told us that he drinks alcohol, gets intoxicated, takes drugs and medicines (abuse of prescription drugs). I desired a husband who would love me and keep me happy. But how can that be possible? I did not have that husband in my destiny." (Young woman)*

Subservience and modesty were described as two parameters that are expected from a "good woman" in a marriage. Most married men and almost half as many married women endorsed similar beliefs that women should obey" the husband and maintain the honour (izzat) of the family by demonstrating modest behaviour in terms of clothing (including veiling) and restricted social interactions. One woman stated, *"She (a woman experiencing violence) must say to him (husband) that I will listen to you and obey whatever you will say. If she will argue about right and wrong with him, then you can understand the outcome of such an argument."* As such, there was an agreement between men and women respondents that any deviation from these norms was said to warrant "discipline and control" towards the woman. Further, men reported that violence in such situations was "justified" in order to keep the woman "beneath her husband". Violence however was also reported when women met the normative expectations. One woman stated, *"Although I used to wear a veil, he was habitual of beating me. He thought that my wife should not see anyone (be seen by anyone), should not talk to anyone."*

*"Violence is justified when they (wives) are wrong. And when are they considered wrong? When they speak in front of someone. Like when someone (guest) is sitting. If they (women) insist on something (they want). If they are insistent somewhere (asking for things) when they should not be." (Young man)*

*"Abuse is needed when the woman is wrong in her behaviour for example talking a lot on the phone or something like that. She needs to be kept under control. If she is talking to some other man. The man (husband) must ask what is your relationship with him and why are you talking to him. Visiting the neighbouring homes very frequently. Going to their homes without any purpose and casually just like that. Based on such behaviour, women need to be controlled." (Young man)*

Young men, women, and community health workers stated in agreement that the woman's expected role in the marriage is to keep the family together, even in situations of conflict or violence. Many women described this as a reason to stay in a violent marriage. It was also described as the reason why a second marriage or divorce is considered a "flaw" in the woman's ability to maintain cohesion without complaints.

*"Our society believes that a good woman is one who suffers the crime of a household and does not tell anyone outside about the crime committed by her husband and in-laws. The situation in our society is still that a woman who suffers from crime is considered a good woman." (Community health worker)*

*"A good woman is patient with everything. She shouldn't fight with her mother-in-law or with the wife of her brother-in-law or if the brother-in-law says something... she should...the result of patience will definitely be sweet. A good woman is someone who, no matter how much her husband scolds or beats her, should feel that she will one day get the fruit of good deeds." (Young woman)*

Most male respondents stated that women are obligated to have sex with their married partners and that refusal of sex is an indicator of infidelity. Many women stated that sex must be provided in order to avoid conflict. Many participants including CHWs and government officials described sexual coercion as wrong, but the lack of consent was not viewed as an anomaly. Many male respondents reported feeling entitled to sex, highlighting broader community perceptions. One man

stated, *"Look, the main thing in life is sex. You understood, right? I continue to have sexual desire fulfilled and I continue to work well also. Now, I don't have any tension, no matter whatever I do. And my wife is also very nice. She is not very demanding. She doesn't say get me this, get me that, she doesn't say all that."*

*"Violence shouldn't happen. It is wrong. Girls go away with boys; mistakes [sexual coercion] happen to girls at an early age." (Community health worker)*

*"I tell him that I don't have any interest (in sex), so stay away from me. He (the husband) gets very irritated by my conduct. Then he creates extreme conflicts, which go on the whole night; neither he sleeps himself nor does he let me sleep. I have now avoided speaking on this matter also, because it causes extreme conflicts, so it is better to tolerate it (demand for sex) silently. It is the best thing." (Young woman)*

*"Forceful sex between husband and wife is also rape. But even if it happens here, no one speaks, if a husband makes a forceful sexual relationship with his wife after drinking alcohol, then the woman does not file a rape case here. It happens in cities; it does not work here." (Community health worker)*

Many participants described having a preference for sons. This was partly due to the norms of providing a dowry when the daughter gets married, which results in financial pressure on the natal family. Other reasons included lineage and the perceived ability of sons to join work at a young age.

*"After that, he had three daughters and he did not have any sons. Then, due to the tension because of this, his mother… then he started drinking alcohol in the daytime. And because of that tension, the environment in the family became negative, because of the fact that he had three daughters, moreover, the family is poor and he also drinks alcohol, he does labour work, how will he be able to raise and take care of his children?" (Young woman)*

While a preference for sons persisted, an emerging pattern suggested that daughters were desired and even better treated if the woman had already had a son. For instance, one woman stated her sister-in-law, *"I have a son, now (God) give me a girl, then I will get the operation (sterilization)."* Similarly, another woman spoke of her husband, *"No, he (husband) never made any differences in both of them (son and daughter). He loves them very much."*

**Norms of masculinity.** Male participants were asked about their identity as a man and the roles and experiences of men in the family and community. Key sub-themes that emerged included norms of men as caregivers, dominance, sexual insecurity, and alcohol use. Men's primary responsibility was described as providing for the family and decision-making. Most women who described their experiences of violence also described the husband's unemployment as a major source of conflict. In such situations, women often took up jobs including sex work but were found to have little to no access to the income as the money they earned was often spent on the husband's alcohol use.

*"If a person always spends earned money on himself (drinking), the children at home are hungry. In such a case, the women have no fault as she has to feed the child, is she wrong? Absolutely not. What do you expect from a woman? She has to feed the family. What will she do? Either she will do prostitution or do home cleaning work in other people's houses to earn money. If a woman is going for home cleaning in another's house, then many men have wrong intentions. Not all men are the same... So, there are many men who have different thinking of a different person in a different house. So, addiction is the main reason that forces a woman to adopt the wrong way to earn." (Young man)*

*"The main factor which is responsible for increasing violence is addiction. Be it an addiction to drugs, alcohol or any type of addiction. Psychologically, sometimes the patient feels pleasure to inflict pain in others. This is also a type of addiction [to violence]." (Local government official)*

Alcohol use was often linked with sexual insecurity. Many men described feeling inadequate in fulfilling their sexual roles within the marriage. This commonly led to the unprescribed use of erectile dysfunction medication, licit and illicit opiates (seen as sex-enhancing) and alcohol as a way of asserting sexual dominance, manifesting in sexual violence and marital rape. Many men described that harmful alcohol use resulted from financial and family-related stressors. Women did not report any reasons that they thought associated with their husband's alcohol use.

*"Many men have thought that my wife is interested in or involved with some other man. Such a man finds himself help-less. If he is not able to satisfy a woman. I mean, if he is a man who is really not physically strong to be able to sexually satisfy his wife. When the feeling of inferiority is always present in mind, he is provoked to get drunk and tries to have sex with his wife as much as possible. This is visible everywhere. Many come [to a pharmacy] for tablets (erectile dys-function drugs) for this problem. This is a common problem." (Young man)*

**Normalisation of violence.** Young women reported that there are few social or legal consequences to disincentivise violence. This contributed to a normalisation of violence and ultimately disempowered survivors from seeking help. Partly it was due to an acceptance by the victim and society that violence is an inevitable or unavoidable aspect of marriage and that it is a private matter to be resolved within the household. Some respondents who identified as upper-caste, also considered violence as normal behaviour within marginalised caste groups that is not seen among dominant caste categories.

*"Yes, they (abused women) tell me that husbands are drunk and do not listen to me (the wife), sometimes they beat children, sometimes they start beating me (the wife). This is very common. They say the same thing in general. Mostly. But there is no record nor FIR for such cases [First Incident Reports (FIR) are the initial information recorded by a police officer given either by the aggrieved person or any other person about the commission of an alleged offence]. Because women are focused on having to run the family, they are engaged in adjustment, and in such adjusting, 10 years pass like this. Then the children grow up and how can they (the woman) then leave." (Local government official)*

*"The panchayat sits in both the in-laws and the parent's house (peehar) place of the girl. Sometimes, if their thinking is evolved, the panchayat (village governing body) also plays a neutral role. It is very rare that the panchayat supports the girl." (Local government official)*

*"Suppose physical violence happens here and the man is beating his wife badly, then nobody can say anything to her, he can continue beating his wife however he wants… If she gets angry and goes to her father's house, then she never comes back, then she forgets her husband. But if she stays with him, then she doesn't say anything to him. I myself fail to understand. I have asked my mother many times. She replies by saying that these are our traditions. We cannot say anything to the husband, no matter what he does. We don't leave the husband." (Young woman)*

Across groups, normalisation also occurred through a general sense that there are "good" and "bad" victims. This is linked to the norms of what a "good" woman should do. Furthermore, young women respondents described that available source of support, such as the police, social services including *sathins*, or community members responded to requests for help by advising the victim to alter their behaviour in order to placate their abusers or by actively supporting the abuser.

*"If something has happened with any girl, and her family members (who commit violence) have given some money to the police, they (police) will refuse the fact, even if they have seen it with their own eyes. This is the matter. As far as the panchayat is concerned… [they say that] the members belong to our neighbourhood, the women are not allowed to speak-up here." (Young woman)*

*"…what is the use of reporting? Nobody can do anything to them…These are the families who make mistakes… despite the families making mistakes, the police let them go after taking the money. They don't do justice for her. If justice is done, I think the woman will definitely go there [to the police]. That's the biggest reason. Because no one listens to the woman. If she is well listened to and understood, the woman will not hold back." (Young woman)*

**Theme 2. Isolation and control**

The normative expectations described above worked in concert with several mechanisms to severely isolate young married women and deny them sources of emotional and material support. Any deviation from the norms would led to increased isolation of the survivor through various means of discipline, surveillance and control of movement; the failure to establish such control could also lead to further violence. Key sub-themes that emerged were a) technology-related control, b) monitoring movement, c) social interactions, and d) modesty. Young women described not being permitted to travel to local towns or markets, getting into trouble for interacting with men and being restricted from accessing technological means of communication such as through telephones or via social media applications such as Facebook or Whatsapp. Women also expressed concerns about being monitored through their social media activities.

*"Whenever I talk to my grandmother on the phone, he (husband) doesn't let me talk to her. Earlier, I had a phone, my mother and all my family had given a phone to me. They told me that if he doesn't let you talk to anyone, so you can talk from this phone. But he even broke that phone. He said that I will not let you talk now. He broke my phone. He does not let me talk to anyone nor does he listen to me." (Young woman)*

These restrictions were justified by a general perception reported commonly by most young male respondents that society is unsafe for women and that social constraints were effective measures of protecting a woman's modesty and honour (*izzat*).

*"They feel the outside environment does not suit them (women); keep girls at home so that nothing wrong happens to them." (Young man)*

Control and isolation also occurred when the husband feared infidelity or had feelings of jealousy. Most young women reported that such fears or feelings of jealousy were unfounded; however, their attempts to prove otherwise did not de-escalate violence in most situations.

*"Yes, I remember that, because the son of my mama (maternal uncle) came. He was lying down outside under the sunlight, so I told him to go in and sleep there [in the shade]…why should he sleep under the sun? I didn't say anything inappropriate to him. He (my husband) was watching me and how I was talking to him. Then he started beating me. He beat me there near the water tanker. I was going to bring the pitcher of water. He came behind me and beat me out in the open in the village. Then I got angry and started crying after getting home. I said, if I have made a mistake, you can even kill me inside (the home). If I didn't do anything wrong, why did he beat me?" (Young woman)*

These restrictions left many women feeling isolated and invisible. Several female participants expressed a deep sense of hopelessness at being unsupported and alone in the world after marriage. Marriages in Sri Ganganagar are often patrilocal, requiring the bride to move to a distant village where the husband resides, which can lead to increased feelings of isolation due to physical distance from the natal family.

## Theme 3. Subjective experience of violence

Many women respondents experienced or witnessed multiple forms of violence. These included incidents of physical, psychological/emotional, financial, and sexual violence that were mostly perpetrated by husbands, though in-laws and natal family members were also implicated.

Two themes emerged when we looked at how women coped with violence: a) de-escalating and preserving relationships, and b) agency and support. Key sub-themes within de-escalating and preserving relationships included responsibility for cohesion and self-silencing. This is interlinked with the norms around marriage, reproduction, and sexuality. Women often stayed in violent marriages to comply or because they had internalised these expectations. Furthermore, speaking about violence and conflict often resulted in further escalation and disharmony perpetrated not only by the husband but the in-laws as well. As such, most women found it better to silence themselves and passively accept violence. Most men did not report an understanding of why women silence themselves or do not seek help. A young man stated, *"But in our village here, the husbands injure their wife's heads completely, and despite of that, they don't go the police station also. They say that he is my husband. I fail to understand why that is so."*

*"He beats me. He keeps a stick inside [the house]. Whenever he gets to know anything (about the wife that concerns him), he drinks alcohol from somewhere outside, then he fights and then goes somewhere else (outside the house). He tells me that it is my fault, even if it is a very small thing. He fights about things like there being too much salt on the vegetables; that the vegetables are very spicy. He starts fighting only over such small things. Researcher: What do you do then? I cry, what else can I do. Researcher: How do you stop him? I don't reply to him and don't say anything. I just keep quiet and go to sleep. Then he eats the food himself and goes to sleep." (Young woman)*

Key sub-themes that emerged within agency and support included *maayka* (natal home) as a refuge, self-reliance, and retaliation. Young women described natal home as a place for temporary shelter that a woman experiencing violence could go to. Often at this point, *bicholas* (matchmakers) and in-laws would visit to bring the daughter-in-law back to her marital home, mostly on the assurance that the husband will improve. The women's ability to access the natal home was however contingent on whether the natal family was supportive or financially stable. Many women from low-income families said that they could never leave an abusive marriage as they would be a burden on their natal family. One woman stated, *"We, the girls of poor families, think like, if we will go and tell our mother (about the violence), still the problem will not decline, it will only progress further. What is the benefit if such matter progresses further? Ultimately, I will need to live there only (at the in-laws)."*

*"If we talk about seeking help… if I ask for help, I will go and stay there [at my parents' home]. They help in the way that my parents come and take me with them and make me stay with them. After that, either my mother-in-law, father-in-law or husband will go there along with the matchmaker as the mediator. Then they will say, "it has been very long since the girl has been living with you, now you send her [back to the marital home]. Now we will not say or do anything to her. There are children [to take care of]. How long will you make her sit with you?" It goes on in this way and they bring me back home." (Young woman)*

*"As such, if a man sexually harasses a woman and if the panchayat gets involved, then the panchayat people say that the boy is wrong. They also abuse him but in the end they will talk to the woman about the honour of the family and force her to end the case. They say "we will make the boy apologise." (Local government official)*

Other women described developing financial self-reliance by taking up work outside the home as a means of coping with violence, particularly financial violence wherein the husband denied the means to look after the children and family. As

described earlier, this income however was often not accessible to the women who earned it. Furthermore, a few women described retaliating against violence with violence.

*"He didn't ask me about the matter and started beating me. Then I got angry and hit my mother-in-law with two slippers. I said, "this all is happening due to you, and now you are not saving me from beatings by him." Then I also hit my mother-in-law [Laughed]." (Young woman)*

We also identified three interrelated themes that prevented women from seeking help: negative experience of disclosure, high threshold for violence within marriage, and duty towards family. Negative experience of disclosure was characterised mainly by the fear of shame and criticism that would arise from family and community, lack of faith in health care providers perpetuated either through personal experience or that of others, and fear of escalation of violence by the perpetrator. Young men emphasised that a woman experiencing violence by their husbands should leave the marriage but offered limited insight into barriers for help-seeking and implications for women.

Violence was also often not reported immediately and only after it became severe. We refer to this as the threshold for violence. This was interrelated with having duties towards children and family. Young women described not seeking help either because they did not have the financial and social means to leave the family or were concerned about their child's wellbeing and future if they were to leave the marriage.

*"A woman never takes any steps too quickly. The woman does not take a step out of fear of disrepute and she gets worried about her children. If she thinks, "I can leave the children..." but a woman doesn't want to leave her children. She can get a divorce and take the wrong step but she doesn't do it because… if she has a daughter, she will think that... if the daughter's rishta (marriage fixing) is done, they (prospective in-laws) will say that her mother has done this, who knows what she (daughter) will be like. Due to this fear, she stops and does not take any steps." (Young woman)*

*"Now, if I am surviving and working, I am doing this all for my children. Then the issue is that the children come back home, they demand something and some other things. I think I will die; I don't want to live anymore. I have gone away once or twice from the home thinking that I will die for sure since he harasses me. I said, "I am living only for my children, not for you." (Young woman)*

## Discussion

The findings of the study highlight GBV as a prevailing public health issue that affects young women in rural Rajasthan. Gender-based violence in rural Rajasthan operates along a continuum—from social conditions such as hegemonic norms of masculinity, constrained agency, and expectations around marriage, to mechanisms of discipline and control including social isolation and monitoring, and finally to overt forms of physical, sexual, and emotional violence. These findings underscore the need to understand GBV as embedded within broader patriarchal kinship structures that shape gendered expectations, authority, and control within marriage, and suggest that effective interventions must engage with these relational and multi-level dynamics.

The findings suggest that these systemic dynamics are deeply shaped by kinship ideologies that organize expectations of marriage, duty, and relational roles [11]. These ideologies produced similarities in how men and women understood and justified violence within marital relationships. At the same time, women were not uniformly passive, and instances of negotiation, reinterpretation, and subtle forms of resistance were evident in their accounts. However, the broad convergence in perspectives across genders underscores the strength of hegemonic patriarchal norms in shaping what is seen as acceptable or inevitable within marriage. This pattern resonates with ethnographic work from North India that shows how gendered norms are internalized and reproduced across social actors, even while leaving room for alternative moral, expressive spaces and agency [32–34].

In terms of attitudes, most participants stated that GBV is wrong and unacceptable, however, there was a wide endorsement of beliefs among married men and women that women should be modest in their appearance, practice veiling, be subservient, restrict their movement for their own safety, and be solely responsible for domestic chores and child rearing. Anthropological studies from the northern states in India highlight how women are socially invisiblised not only within the household, but within the larger community, by limiting their self-expression in public contexts and restricting interaction with dominant-caste senior men [35]. At the same time, ethnographic work from North India highlights that women's voices are not entirely silenced, but may be expressed in alternative or less visible spaces, including through collective action, narrative, and everyday forms of negotiation [36]. However, the responses in this study—elicited in formal interview settings—largely reflected dominant and publicly sanctioned norms, underscoring the strength of hegemonic expectations. In our study, these beliefs were held across genders, and women who did not associate with these norms were considered "bad" women that deserve violence. This is in line with other quantitative findings from the northern states of India, that women who are disobedient are deserving of verbal and physical abuse from their husbands, such as in situations where a woman does not cook properly or goes out without informing the husband [37,38]. The problematic nature of such beliefs shows how violence becomes socially sanctioned when women deviate from expected gender norms, particularly those governing obedience, mobility, and domestic roles. The irony, however, is that women who are less empowered, economically and educationally, are also at high risk of experiencing violence [39].

Future GBV programming must situate the changing landscape of women's empowerment in India within the intersectionality of class, caste and gender, considering that the findings reveal that one size does not fit all. These findings must be situated within the intersection of caste, class, and gender in rural Rajasthan, where hierarchies of labour, land ownership, and social status shape both exposure to violence and access to support. While this study was not designed to systematically compare caste groups, the patterns observed suggest that marginalised communities may experience compounded vulnerabilities linked to economic precarity and social exclusion [27,40]. Further ethnographic work is needed to more fully understand how these dynamics vary across caste and community contexts.

At the same time, the regulation of women's sexuality and mobility observed in this study resonates with broader literature on honour (izzat) and sexual purity in North India. Some scholars have argued that processes of social mobility and modernity are associated with the adoption or reinforcement of more restrictive, brahmanical norms governing women's sexuality across caste group [32–34]. However, the specific context of Sri Ganganagar—characterized by a relatively recent settlement history and a demographic composition in which dominant caste groups are not uniformly upper-caste Hindu—suggests that these dynamics may not map neatly onto classical formulations of brahmanical patriarchy. Rather, concerns around family honour and women's conduct appear to be locally configured, while still producing similar constraints on women's autonomy.

It was widely believed by married men and women as well as community health workers and government officials that men should bear financial responsibilities as part of being a "good husband". This normative pressure in combination with very few stable job opportunities in the area led to frustration and drinking as a means of coping. Sri Ganganagar is primarily an agricultural district and many men involved in agricultural labour work engage with it during sowing and harvesting period as it is often provided by employers as part of their compensation, which ranges across two to three months in a year. During the rest of the year, job opportunities are limited to informal or irregular labour tasks, and many men, particularly from minority caste groups, find themselves in a position where the survival of the family becomes difficult [41]. Even long-term labourers, primarily from marginalised (SC/ST) communities, often engage in unfree labour due to lack of other opportunities, access to land, and assured employment during agricultural season [42]. Previous studies show that when employment, which is contextually considered as a marker of being a "man", is threatened, alcohol use increases [43]. The complex interaction of markers of masculinity, GBV, and alcohol and substance use warrants further studies in order to develop effective prevention programs. These findings also point to the ways in which patriarchal kinship systems place normative pressures on men, particularly in relation to provision and authority within the household. When these

expectations cannot be met, they may generate frustration and contribute to patterns of alcohol use and violence. This underscores that patriarchal systems not only constrain women but also shape men's experiences in ways that may perpetuate cycles of harm [44].

The strength of these kinship ideologies was evident in the normalisation of violence within marriage, which was widely described as an expected part of married life and one for which long-term support does not exist. Sexual violence or marital rape in particular, was highly normalised. India is one of the 36 countries where marital rape is not a crime [45]. The Domestic Violence Act recognises sexual violence within marriage, but does not criminalise it, and instead offers civil remedies such as separation and divorce [17]. Divorce or separation, however, is uncommon. Our findings are in line with research from South Asia which shows that divorce is often not sought because birth family support may not exist or the option of earning income is not available [46]. Violence was also more commonly observed in women who did not give birth to sons. While many women reported violence, only one woman in the study sample was found to seek divorce as a response.

Although examination of formal support systems was not a primary focus of this study, participants' accounts point to significant gaps in the accessibility and responsiveness of available services, which are themselves shaped by prevailing social norms. Community health workers and government officials endorsed the belief that women experiencing violence should seek help however also reiterated that often systems that are put in place to provide support retraumatise the survivors by minimising their experiences of GBV. Women were expected to maintain harmony in the family even in the face of conflict and those unable to do so are shamed, criticised, or gossiped about. It was also widely endorsed that women do not leave abusive relationships because of lack of social security for children and associated stigma. Community health officials described hesitation to intervene as IPV was often described as a "private matter" and intervening was considered unsafe. At the same time, field observations by members of the research team (AB) indicated that some institutional responses, such as the One Stop Centre in the study area, were functioning and accessed by women, with the availability of counselling, legal, and police support. This suggests that while structural and social barriers to help-seeking remain significant, existing services may offer important points of support when accessible and trusted. This also highlights that establishing the safety of community health workers that belong to the very same patriarchal communities is a critical part of their training and capacity building to address GBV. The rape of Bhavari Devi, a *sathin* in Rajasthan, who intervened with a child marriage is a strong example of the lack of safety measures for health workers [47]. *Sathin*, which translates to friend, was the Rajasthan state government's Women's Development Programme (WDP), which notably struggled to empower women as workers [48]. Additionally, health workers described limited competency to screen and identify signs of violence.

There are several research, political, and policy implications of our findings. These implications must be understood within the specific context of rural districts such as Sri Ganganagar, where geographic distance, agricultural labour relations, and reliance on institutions such as the family (mediated through bicholas or marriage brokers) and panchayats, both formal and informal, shape both the experience of violence and the possibilities for intervention. These contextual features distinguish rural settings from the predominantly urban contexts in which much GBV research and programming has been developed, where service access, resources, mobility, and institutional engagement may differ significantly.

Young women between the ages of 14–25 experience marriage, childbirth, and violence on a continuum. Despite official statistics that suggest that child marriage is no longer as prevalent as in past decades, many of our participants reported being married, having sex, and getting pregnant soon after menarche [49].

Policymakers and implementors must focus on strengthening the implementation of existing child-protection policies, and work to change the community's perception of child marriage as a human rights violation. Community awareness drives and use of role models in communication could be possible tools. A key challenge in child marriage prevention is in part caused by the legal design where child marriage is recognised per se, unless nullified by either of the parties involved [50]. To counter this, National Commission of Women proposed The Draft Compulsory Registration of Marriage

Bill, though it continues to be in between legislative proceedings [51,52]. There is a need for compulsory age verification at the time of marriage and combat the pervasive use of fake birth certificates [51]. National campaigns such as "Beti Bachao, Beti Padhao" (Save Girls, Educate Girls) have led to positive outcomes over time in Haryana, Rajasthan, and tribal regions of Jharkhand, yet key issues persist in changing deep-rooted patriarchal beliefs and maintaining adequate monitoring [53].

There are very few safe and non-judgemental avenues to seek help for women survivors of GBV or IPV. Programs must focus on building capacity at all levels in health centres, one-stop centres (One Stop Centres are government centres designated to provide integrated support and assistance to women affected by violence and in distress, both in private and public spaces under one roof and facilitate immediate, emergency and nonemergency access to a range of services including medical, legal, temporary shelter, police assistance, psychological and counselling support to fight against any forms of violence against women), *panchayats*, and hospitals to improve screening, referral mechanisms, and overall support, especially for rural women. Other family and community members including mothers-in-law, father-in-law, and *bicholas* play a unique role in the married couple's life. Sensitisation of these members would be important in future interventions including when and how to intervene, and how to provide emotional and logistical support to survivors.

Interventions in this context must engage with, rather than attempt to bypass, the relational structures within which violence occurs. As suggested in ethnographic work on anti-violence counselling in India, addressing GBV often involves working within kinship systems to reorder relationships, rather than assuming that women are able to exit them entirely [11,54]. This underscores the importance of engaging not only men, but also broader family and community networks, in efforts to shift norms and support survivors. Couple-based and family-oriented approaches may therefore offer promising avenues for intervention.

Our study has a few limitations. The data was collected at a singular-time point and hence it is difficult to say how gender-based violence and associated concerns persist and/or resolve over time. Consistent with the profile of the community, the study population was predominantly limited to Hindu and Sikh populations and we did not have enough context on experiences of other minority populations in India such as Muslims and Christians. Finally, our data does not include perceptions and attitudes of family members who play a key role in mediation of marriage as well as resolution of conflicts, such as in-laws, natal family members, and *bicholas*. On the other hand, our study also has several strengths. The study includes various stakeholders and highlights the subjective experiences of GBV perpetration and victimisation which have largely previously been explored quantitatively. By foregrounding the perspectives of participants in their own words, this study contributes contextually grounded insights into how violence is experienced, justified, and negotiated within everyday life in rural Rajasthan. Our findings highlight that while many policies and programs are in put in place to protect women, women in rural Rajasthan remain vulnerable in ways largely unchanged for decades.

## Conclusion

Gender-based violence is multifaceted and stems from disempowerment and control established early in a woman's life. Programs and policies aimed at violence prevention must focus on markers such as early marriage and son preference along with establishing safe support networks for women in order to create a healthier society.

## Acknowledgments

The authors would like to acknowledge the contribution of Dr Vikram Patel, Paul Farmer Professor and Chair of Global Health and Social Medicine, Harvard Medical School as the Principal Investigator of the research project. We would like to acknowledge the contributions of Dr Prerana Pandia and Dr Amit Dhage towards data collection with the study participants and data analysis.

## Author contributions

**Conceptualization:** Aneel Brar, Urvita Bhatia, Abhijit Nadkarni.

**Formal analysis:** Soumya Singh, Aneel Brar.

**Funding acquisition:** Aneel Brar, Abhijit Nadkarni.

**Investigation:** Aneel Brar.

**Methodology:** Aneel Brar, Urvita Bhatia, Devika Gupta, Abhijit Nadkarni.

**Project administration:** Soumya Singh.

**Supervision:** Aneel Brar, Abhijit Nadkarni, Urvita Bhatia, Devika Gupta.

**Writing – original draft:** Soumya Singh.

**Writing – review & editing:** Aneel Brar, Urvita Bhatia, Devika Gupta, Abhijit Nadkarni.

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
