## [Decision Letter · Decision Letter 1]

11 Feb 2026

PGPH-D-25-01450

Multi-stakeholder Perspectives on Gender-based Violence in Marital Relationships: A Qualitative Study in Rural India

Dear Dr. Nadkarni,

Thank you for submitting your manuscript to PLOS Global Public Health. After careful consideration, we feel that it has merit but does not fully meet PLOS Global Public Health’s publication criteria as it currently stands. Therefore, we invite you to submit a revised version of the manuscript that addresses the points raised during the review process.

The manuscript has been assessed by two reviewers and their comments are available below. Please review their reports and make the appropriate revisions to address any concerns.

We look forward to receiving your revised manuscript.

Kind regards,

Emma Campbell, Ph.D

Staff Editor

Journal Requirements:

1. Please ensure that your Ethics Statement is available in its entirety at the beginning of your Methods section, under a subheading 'Ethics Statement'.

2. In the online submission form, you indicated that “The datasets generated during and/or analyzed during the current study are not publicly available due to reasons of confidentiality but are available from the corresponding author on reasonable request.”

3. Uploaded as supplementary information.

Additional Editor Comments (if provided):

Reviewers' comments:

Reviewer's Responses to Questions

**Comments to the Author**

1. Does this manuscript meet PLOS Global Public Health’s publication criteria? Is the manuscript technically sound, and do the data support the conclusions? The manuscript must describe methodologically and ethically rigorous research with conclusions that are appropriately drawn based on the data presented.

Reviewer #1: Yes

Reviewer #2: Yes

2. Has the statistical analysis been performed appropriately and rigorously?

Reviewer #1: Yes

Reviewer #2: Yes

3. Have the authors made all data underlying the findings in their manuscript fully available (please refer to the Data Availability Statement at the start of the manuscript PDF file)?

Reviewer #1: Yes

Reviewer #2: Yes

4. Is the manuscript presented in an intelligible fashion and written in standard English?

Reviewer #1: Yes

Reviewer #2: Yes

5. Review Comments to the Author

Reviewer #1: I appreciate the authors for conducting this much needed public health research. Minor revisions need to be done before next stage.

Abstract: kindly write a structured abstract with more details on research results, conclusion and recommendations.

Key words: add substance use, masculinity, IPV, GBV, alcoholism, marital rape

Introduction:

Line no 35: instead of underage marriage use the word child marriage and mention child marriage act and POCSO act as well

Line no 36: mention latest NFHS results

Line no 44-49: add dowry harassment, dowry death, honour killing and bride burning data

Add existing dowry harassment act, domestic violence act, of relevant mental health care act

103-110- what was the formal training provided to sathins add details and references

Any vicarious trauma? How was it prevented

248-249: need to reflect whether this statement actually reflects the restriction of woman’s autonomy

Discussion: authors need to strengthen discussion using intersectional feminism lens to write about the plight of married women in India

Health system strengthening: one stop centres are actually a failure model. Please critically evaluate this recommendation and suggest feasible scalable culturally appropriate recommendations such as capacity building of ASHAs, AWWs, ANMs, CHOs. Why AB HWCs, PHC, AWC cannot act as safe spaces for women?

VHSNCs - why can’t they advocate for the rights of women, what are the opportunities

You can also elaborate on challenges faced by ministry of WCD in curbing child marriages and implementing Beti bachao beti padao campaign strategies

Reviewer #2: The paper is well written. The study provides valuable data in an understudied setting. It will benefit from some small tweaks for increased clarity-

In the introduction, the denominator is unclear when you are citing NCB statistics. Please clarify.

The methodology is sound. What was the purpose of the double blind though, as it wasn’t an experimental study? How may hiding the purpose of the study have affected results? It might also be useful to provide some information about your SSI and FGD questions to ensure transparency. Also, what is the definition of violence you are using? does it encompass all kinds of physical, emotional, sexual and financial abuse?

Results and discussion-

I think the theme code ‘marriage as a single blind experiment’ needs to be rephrased. This code, while witty, does not accurately capture what is essential deception and lack of agency for women when making a choice of their partner. An experiment has a purpose- what is it in this case?

Some of the sub themes can be explained, in a table if needed, for clarity.

While data triangulation is helpful, in this instance, it might be helpful to also establish differences in the ways in which violence is viewed across genders.

While you have representation across class and caste, it might be useful to comment on whether your data revealed trends based on intersectional parameters. For instance, did caste disenfranchisement affect norms of masculinity? did brahmanical patriarchy related values affect notions of sexual purity?

Please locate your results in the context of existing research on GBV primarily conducted in urban areas. Are there context-specific differences you noted? Also comment briefly on resources available for help, in lieu of PWDA act or lack of support for marital rape, and the reality of how these operate in the rural context, which helps to put some of the help seeking limitations and gaps into perspective.

It might also be helpful to situate the data using existing theories in India, such as local masculinity conceptualisations, notions of collectivist of honour, or the Chakrabhed cycle of marital and natal family violence by Gupte and More in Padma Deosthali’s 2012 book.

6. PLOS authors have the option to publish the peer review history of their article (what does this mean?). If published, this will include your full peer review and any attached files.

**Do you want your identity to be public for this peer review?** For information about this choice, including consent withdrawal, please see our Privacy Policy.

Reviewer #1: **Yes:** Nancy Angeline Gnanaselvam

Reviewer #2: No

Figure Resubmissions:

---

## [Decision Letter · Decision Letter 1]

10 Apr 2026

Multi-stakeholder Perspectives on Gender-based Violence in Marital Relationships: A Qualitative Study in Rural India

PGPH-D-25-01450R1

Dear Dr. Nadkarni,

We are pleased to inform you that your manuscript 'Multi-stakeholder Perspectives on Gender-based Violence in Marital Relationships: A Qualitative Study in Rural India' has been provisionally accepted for publication in PLOS Global Public Health.

Best regards,

Dr Tanmay Bagade, Ph.D., MS (O&G), MPH, MHM

Academic Editor

Reviewer Comments (if any, and for reference):

Reviewer's Responses to Questions

**Comments to the Author**

1. If the authors have adequately addressed your comments raised in a previous round of review and you feel that this manuscript is now acceptable for publication, you may indicate that here to bypass the “Comments to the Author” section, enter your conflict of interest statement in the “Confidential to Editor” section, and submit your "Accept" recommendation.

Reviewer #2: All comments have been addressed

2. Does this manuscript meet PLOS Global Public Health’s publication criteria? Is the manuscript technically sound, and do the data support the conclusions? The manuscript must describe methodologically and ethically rigorous research with conclusions that are appropriately drawn based on the data presented.

Reviewer #2: Yes

3. Has the statistical analysis been performed appropriately and rigorously?

Reviewer #2: Yes

4. Have the authors made all data underlying the findings in their manuscript fully available (please refer to the Data Availability Statement at the start of the manuscript PDF file)?

Reviewer #2: Yes

5. Is the manuscript presented in an intelligible fashion and written in standard English?

Reviewer #2: Yes

6. Review Comments to the Author

Reviewer #2: I am satisfied with the changes made. The study seems more robust following this round of revisions.

7. PLOS authors have the option to publish the peer review history of their article (what does this mean?). If published, this will include your full peer review and any attached files.

**Do you want your identity to be public for this peer review?** For information about this choice, including consent withdrawal, please see our Privacy Policy.

Reviewer #2: **Yes:** Jagruti Wandrekar
